# Elliptical Orbit Design Based on Air-Breathing Electric Propulsion Technology in Very-Low Earth Orbit Space

**Yuxian Yue \*, Jinyue Geng, Guanhua Feng and Wenhao Li \***

Institute of Mechanics, Chinese Academy of Sciences, Beijing 100190, China; gengjy@imech.ac.cn (J.G.); fengguanhua@imech.ac.cn (G.F.)

\* Correspondence: sy1715318@buaa.edu.cn (Y.Y.); liwenhao@imech.ac.cn (W.L.); Tel.: +86-13240308959 (Y.Y.)

**Abstract:** Very-low Earth orbit (VLEO) space below 200 km is essential for high-quality communications and near-Earth space environment detection. Due to the significant atmospheric drag, orbital maintenance is required for spacecraft staying here. Based on air-breathing electric propulsion (ABEP) technology, this paper analyzed the orbital boundary conditions of the spacecraft under the constraints of parameters including slenderness ratio, thrust-to-power ratio, drag coefficient, and effective specific impulse. The energy balance is the key constraint for low VLEO orbits, which is determined by the drag coefficient, slenderness ratio, and thrust-to-power ratio. Under the existing technical conditions, the lowest circular orbit (along the terminator) is about 170 km. An elliptical orbital flight scheme is also analyzed to reach a 150 km perigee. A half-period control method was proposed based on the on–off control method for the elliptical orbit, which could enable the spacecraft to maintain a stable 150–250 km elliptical orbit.

**Keywords:** very low Earth orbit; air-breathing electric propulsion; elliptical orbit; control method

## 1. Introduction

The very-low Earth orbit space (especially the space below 200 km) is of great significance for high-quality Earth observation and communication because of its short distance to the Earth's surface [1–3]. Lower orbital altitudes and longer orbit maintenance can significantly improve payload performance and cost savings. For example, the launch cost of a spacecraft on a 200 km orbit can be reduced by 10–50% compared to 400 km, while the observation resolution or communication quality to the Earth can be increased by 2–4 times [4]. In addition, the very-low orbit space near 150 km is the main region of the ionosphere F1 layer (130–210 km), which is an ideal place for space science measurements and experiments. In situ measurements of the upper atmosphere, ionosphere, and Earth's gravity field can be carried out there, providing conditions for geological survey and natural disaster monitoring through the measurement of ionospheric parameters [5–7]. In recent years, Europe and Japan have launched the GOCE satellite (ESA, 250–300 km) and Tsubame satellite (SLATS) (JAXA, 180–250 km) to VLEO space respectively, bringing us environment exploration and technical verification [8–10]. Lixing-1 satellite launched by CAS (China Academy of Science) flew around the Earth 36 times (about 52 h) at an altitude of 109–150 km, collecting atmospheric resistance data in this area [11]. However, there are still gaps in long-term in situ exploration below 200 km.

There are several complex perturbations in VLEO space including atmospheric resistance and high-order earth non-spherical perturbations. Atmospheric resistance is the most significant perturbation, and the non-thrust spacecraft can only be maintained for a short time under the action of atmospheric resistance. A spacecraft with 1 m$^2$ cross-sectional and 4 tons mass can only last 50–150 h at an altitude of 150–180 km (Figure 1).

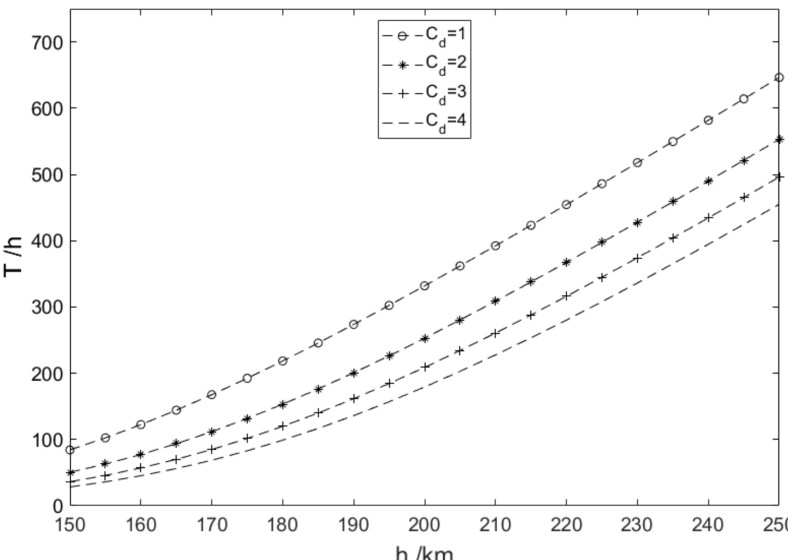

**Figure 1.** Approximate orbital maintenance time of a spacecraft without thrust under different aerodynamic drag coefficients $C_d$.

To maintain orbit in the long term, the spacecraft is required for long-lasting on-orbit propulsion. The existing VLEO spacecraft such as GOCE and Swallow use electric propulsion (chemical working fluid), and their propellant mass accounts for a significant proportion (usually >50%). To reduce the mass of the propellant, the concept of air-breathing electric propulsion (ABEP) was proposed. ABEP is designed to collect the air composition in the high atmosphere and use it as electric propulsion working fluid, drastically reducing the mass of the carrying propellant.

In 2003, JAXA proposed the concept of ABEP and carried out a series of studies [12–15]. ESA designed a suction device suitable for ABEP thrusters in 2007, analyzing the feasibility of suction in the VLEO area (180–250 km) [16]. Romano et al. systematically analyzed the atmospheric properties and drag of the VLEO region based on the key parameters of ABEP spacecraft [17–20]. In 2017, Jackson et al., and Peng Zheng et al. respectively designed the ABEP system used in Cubesat and carried out simulation verification, which provided a reference for the design of the ABEP system for larger spacecraft [21,22].

In practice, SITAEL's space team first tested a complete RAM-EP system in a representative environment (the Aether project) [23,24]. In this project, Andreussi et al. studied the feasibility of using Hall-effect thrusters in air-breathing electric propulsion and gave the scope of application of high-thrust-power ratio thrusters by a series of experiments [25,26]. Andreussi et al. also made a comprehensive review of the main research and development efforts on the ABEP technology [27]. Although there are some technical issues to be solved, the previous works still show that ABEP technology can be applied to engineering in recent years.

In terms of atmosphere resistance analysis and reduction, Tisaev et al. systematically analyzed the influence of air-breathing electric propulsion spacecraft performance on the feasible flight area of VLEO based on the characteristics of atmosphere resistance [28]. In addition, there are studies by Jiang Y et al. and Andrews S et al. on the aerodynamic shape design and drag reduction control strategy of VLEO spacecraft [29,30].

Based on the listing research on air-breathing electric propulsion and atmospheric resistance, this paper proposes an elliptical orbital flight scheme in VLEO space below 200 km. First, the working fluid balance and energy balance of ABEP in VLEO space are analyzed based on four key parameters: slenderness ratio, thrust-to-power ratio, drag coefficient, and Effective specific impulse. The energy balance was found hard to reach, which limits ABEP's orbital flight in VLEO space lower than 200 km. Elliptical orbital flight schemes were proposed and analyzed, which reduces the energy gap. Then, the orbital parameter constraints of the ABEP spacecraft are calculated according to the slenderness

ratio, thrust-to-power ratio, and drag coefficient. Finally, the control method of the elliptical orbital flight is given.

Firstly, based on the analysis of the upper atmosphere, the performance of the air-breathing electric thruster, and the shape of the spacecraft, it was determined that the main factor limiting the maintenance of the orbit of the air-breathing electric propulsion vehicle at an orbital altitude of 150 km is the energy input. Then, according to the key parameters of the spacecraft (slenderness ratio, thrust power ratio, and drag coefficient), the elliptical orbit constraints of the ABEP spacecraft are analyzed, and the control method is given. Finally, the ABEP vehicle elliptical orbital constellation is constructed according to the effective coverage.

## 2. ABEP Flight Scheme Analysis

### 2.1. VLEO Space Environment and ABEP Concept

Unlike other Earth orbit space, the VLEO space (especially below 300 km) has a significant atmospheric composition. According to the MSISE-00 model developed by NASA, the average atmospheric density (average value along the terminator on 21 March 2020) is shown in Figure 2. In this figure, we interrogated the density data for every 10 km altitude and made a continuous numerical fit: $\rho = \rho(h)$. The data is from 100 km to 300 km, which can be found at the NASA database website: https://kauai.ccmc.gsfc.nasa.gov/instantrun/nrlmsis/ (accessed on 25 September 2023). We chose the average value along the terminator because terminator orbits are chosen for ABEP spacecraft, which will be discussed later in Section 2.2.

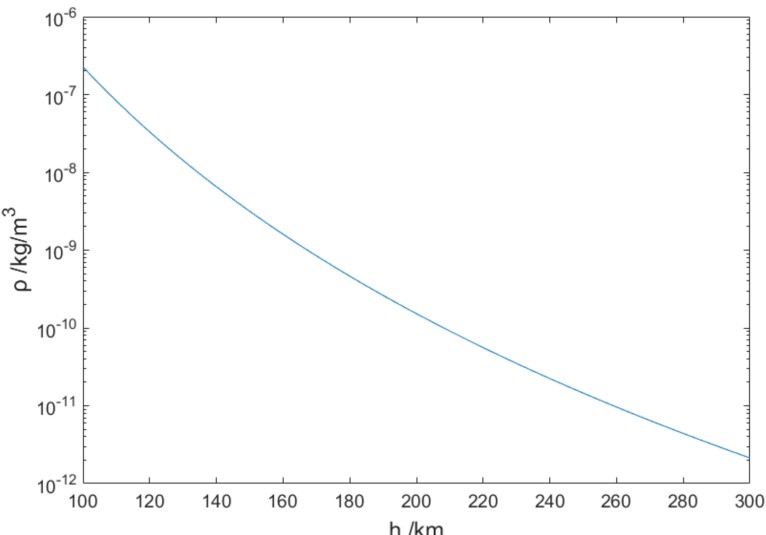

**Figure 2.** Atmospheric density (average value along the terminator) vs. altitude, NASA: MSISE−00 atmospheric model fitted values.

It should be noted that the atmospheric density of the VLEO region varies (generally not more than 40%) with time and position (latitude and longitude), which is mainly caused by the insolation angle and the local low-level atmospheric flow [31,32]. Since this variation is hard to predict, and there is a cumulative effect of the long-term orbital flight, the atmospheric density is considered to be the orbital average value at each altitude. In addition, strong solar activities also have effects on atmospheric density [33], which is low-frequency and difficult to predict, so these episodic events will not be further discussed in this study.

The rarefied atmosphere in VLEO space presents two challenges for spacecraft staying here. The first is the problem of aerodynamic resistance, especially the equilibrium thrust control problem in the case of atmospheric instability. The second is the problem of material corrosion and friction loss, mainly caused by the atomic oxygen in the ionosphere. This

paper focuses on the first problem of aerodynamic resistance. The existing ABEP technology is mainly considered.

ABEP system is usually composed of three main parts: an air intake module, a compression module, and an electric propulsion thruster. The air intake module collects the high relative velocity of incoming lean air and sends it to a compression chamber for compression, which uses passive or active compression methods to compress the lean air to the concentration available for thruster ignition. Finally, electric propulsion thruster uses compressed air as the working fluid for propulsion.

Hall thrusters with high thrust can be used on ABEP spacecraft, which provide a relatively high specific impulse of up to 5000 s and a high thrust-to-power ratio of more than 50 μN/W [34].

*2.2. Dynamics Analysis of ABEP Spacecraft in VLEO Space*

Due to the complex atmospheric environment of VLEO space, the aerodynamic drag is more complex than that of high orbits. Research related to the ionosphere [32] shows that the flow field in this region is between continuous flow and free molecular flow, so it is necessary to consider both continuous flow drag and free molecular flow drag when deciding the drag coefficient. Since the atmosphere parameters are not precise and the drag is closely related to the shape of the spacecraft, the actual drag coefficient of ABEP spacecraft (various aerodynamic shapes) $C_d$ must be measured in situ or by rarefied wind tunnel experiments. The drag coefficient range given in reference [35] is about 1–4.

It is worth noting that in order to obtain maximum power from solar arrays and ensure the stability of the atmospheric environment, a terminator orbit can be chosen for the ABEP spacecraft.

According to classical resistance theory, the aerodynamic drag $F_f$ to which the space-craft is subjected can be written as:

$$F_f = \frac{1}{2}\rho v^2 C_d(b) S_c \tag{1}$$

where $\rho$ is the atmospheric density (acquired by MSISE-00 atmospheric model), $v$ is the relative speed of the spacecraft and the atmosphere (the atmospheric movement velocity is neglected as it is small relatively), and $S_c$ is the effective cross-sectional area of the spacecraft. In order to simplify the complex effects of attitude, this study argues that the main attitude axis of the spacecraft is always parallel to its orbital velocity direction.

For thrusters with fixed specific impulses, the rated thrust $F_t$ is:

$$F_t = \dot{m} g_0 I_{sp} \tag{2}$$

The gas flow taken by ABEP is:

$$\dot{m} = \eta \rho v S_t \tag{3}$$

where $I_{sp}$ is the specific impulse of the thruster, $g_0$ is the acceleration of gravity, $\eta$ is the suction efficiency of the ABEP thruster, and $S_t$ is the aspirated cross-sectional area of the air-breathing electric propulsion. Active suction devices or spiral wave discharge methods can be used to increase the compression ratio and achieve stable operation at lower air pressures. Therefore, the compression ratio limitation is not discussed here.

To ensure the stability of the ABEP spacecraft in orbit, the thrust needs to be greater than the drag on average.

According to (1) and (2), drag and thrust are both proportional to atmospheric density $\rho$. To quantify the equilibrium relationship, define the normalized thruster-to-drag ratio $R_{tf}$ independent of atmospheric density:

$$R_{tf} = \frac{F_t}{F_f} = 2\frac{g_0}{v C_d} \cdot I_{sp} \cdot \eta \cdot \frac{S_t}{S_c} \tag{4}$$

When the value of $R_{\text{tf}}$ is greater than or equal to 1, the working fluid acquired by the air-breathing electric propulsion spacecraft is possible to maintain its flight orbit. Obviously, the working fluid equilibrium under different drag coefficients is decided by the effective specific impulse $I_{\text{sp}} \cdot \eta$ (product of suction efficiency and specific impulse, characterizing the effective specific impulse for incoming flow), aspirated cross-sectional ratio $S_{\text{t}}/S_{\text{c}}$ (ratio of aspirated cross-sectional area and total cross-sectional area) and orbital velocity $v$. As the orbital velocity changes slightly during 100–300 km altitude and can be approximated as a constant value, the thruster-to-drag ratio $R_{\text{tf}}$ is mainly determined by the effective specific impulse $I_{\text{sp}} \cdot \eta$, aspirated cross-section ratio $S_{\text{t}}/S_{\text{c}}$, and drag coefficient $C_{\text{d}}$. Figure 3 shows the normalized thruster-to-drag ratio vs. effective specific impulse and cross-section ratio considering J2 perturbation under different drag coefficients.

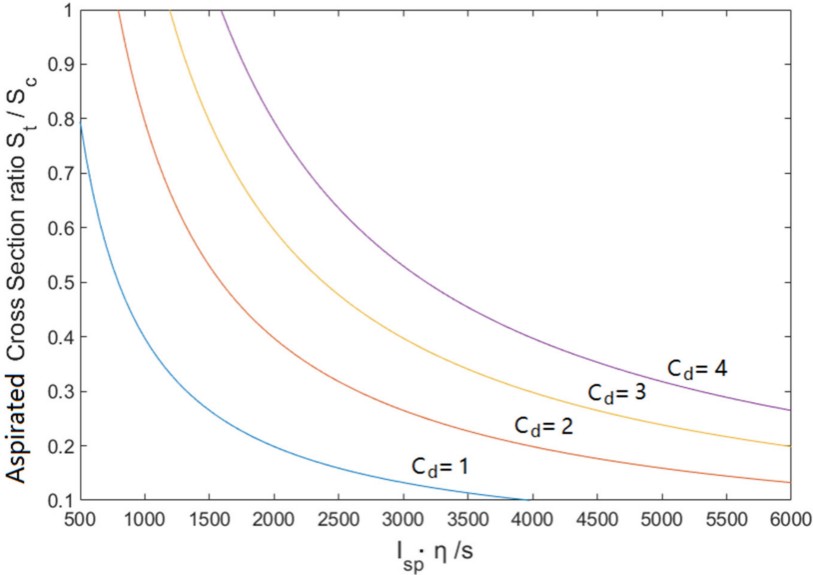

**Figure 3.** The thruster-to-drag ratio is decided by effective specific impulse and aspirated cross-section ratio.

Similarly, the normalized power ratio $R_{\text{io}}$ can be given by the propulsion power $P_{\text{o}}$ and the solar array power $P_{\text{i}}$:

$$P_o = F_t(\min) \cdot v = F_f v = \frac{1}{2}\rho v^3 C_d S_c \tag{5}$$

$$P_i = S_b I_0 k \tag{6}$$

$$R_{io} = \frac{P_i \cdot \varepsilon}{P_o} = 2\frac{I_0 k}{C_d} \cdot \frac{\varepsilon}{v} \cdot \frac{S_b}{S_c} \cdot \frac{1}{\rho v^2} = 2\frac{I_0 k}{C_d} \cdot t_{pr} \cdot \frac{S_b}{S_c} \cdot \frac{1}{\rho v^2} \tag{7}$$

Among them, $S_{\text{b}}$ is the effective area of the solar array, which is generally the side area of the spacecraft (such as the side patch solar arrays of GOCE [9]) or the area of the attached wings (such as the side wing arrays of Tsubame [10]). For the convenience of normalization calculations, this area is collectively referred to herein as the side area. $S_{\text{c}}$ is still the cross-sectional area. $I_0$ is the solar radiation constant, taking 1364 W/m$^2$. $k$ is the solar array conversion efficiency, usually taken as 0.3. $\varepsilon$ is the power proportion of the propulsion system, that is, the ratio of the power required for ABEP to the total input power, and $t_{\text{pr}} = F_{\text{t}}/P_{\text{o}} = \varepsilon/v$ is the thrust-to-power ratio.

When the normalized power ratio $R_{\text{io}}$ is greater than or equal to 1, the energy obtained by the ABEP spacecraft through its own solar arrays can maintain its flight trajectory. The energy balance under different drag coefficients is determined by the orbital height $h$, the thrust-to-power ratio $t_{\text{pr}}$, and the side-to-cross-section ratio $S_{\text{b}}/S_{\text{c}}$, and the relationship is

shown in Figure 4, where the space above the curved surface is the feasible domain of these three parameters.

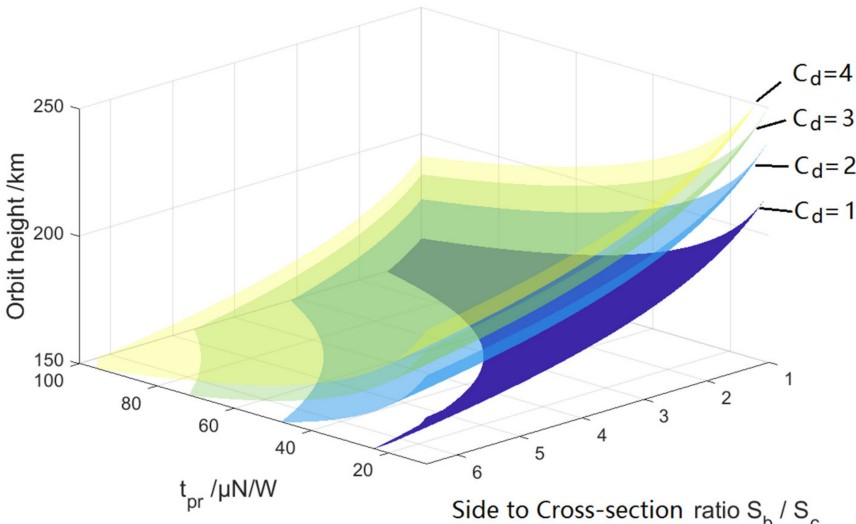

**Figure 4.** The power ratio domain decided by the thrust-to-power ratio, orbit height, and side-to-cross-section section ratio.

Figures 3 and 4 illustrate two balance (working fluid and energy) limits of ABEP spacecraft concerning orbital altitude, effective specific impulse, thrust-to-power ratio, geometric parameters, and drag coefficients.

Since the aerodynamic force makes the spacecraft very unstable at the large slenderness ratio [36], the side-to-cross-section ratio $S_b/S_c$ generally does not exceed 5. The aspirated cross-section ratio $S_t/S_c$ can reach nearly 1 under proper design. The thrust-to-power ratio $t_{pr}$ is typically 20–100 $\mu$N/W, and the effective specific impulse ($I_{sp} \cdot \eta$) is typically 1000–5000 s for current thrusters [34]. Therefore, it can be seen from Figures 3 and 4 that the power ratio is the main constraint for ABEP vehicles in VLEO space below 200 km, while the working fluid requirements can be met easily.

## 3. Elliptical Orbit Parameters Constraints and Control Method

In order to meet the energy requirements which means to improve the power ratio, elliptical VLEO orbits can be considered. The perigee of the orbit is lower than 200 km and there is an orbit section that meets the height requirements of the payload, which is called the mission interval.

At high orbit altitude, the spacecraft is subjected to less air resistance with a lower orbital speed. As the propulsion power is less than the input, energy can be stored during this flight interval, which can be called the energy replenishment interval.

According to the characteristics of the elliptical VLEO orbit, the orbital parameter constraints and control methods are analyzed.

### 3.1. Constraints Analysis of Elliptical Orbit

Firstly, the approximate feasible domain of elliptical orbit parameters is analyzed according to the power ratio described in Section 2. The influence of unsteady changes in $\rho$ is not considered, and the atmospheric density $\rho$ is used as a function of orbital altitude $h$ according to the NASA: MSISE-00 atmospheric model. When the variation of $\rho$ is not large and its mean value is close to the model value, the orbital parameter constraint obtained by theoretical analysis is approximated as the actual constraint.

According to the analysis in Section 2, the constraints on orbital parameters mainly consider the energy balance decided by the power ratio. Since the energy storage module can distribute the stored electrical energy according to the thruster demand, it can be

considered that as long as the energy obtained in an orbital period is greater than the energy consumed, the orbit is theoretically feasible.

In the case of considering the energy balance, it must meet the following:

Therefore, for an elliptical orbit

$$r(t) = \frac{p}{1 - e\cos(\theta(t))}, \tag{8}$$

The integral value (energy per period) can be completed according to (5) and (6):

$$E_o = \int_0^T \frac{1}{2}\rho(h(t))v(h(t))^3 C_d S_c dt \tag{9}$$

$$E_i = \int_0^T S_b I_0 k dt \tag{10}$$

where $T$ is the orbital period, $h$ is the orbital altitude, and $\rho(h(t))$ is the approximate parameter function obtained from the NASA: MSISE-00 atmospheric model.

In the case of considering the energy balance, the constraint must be met:

$$E_i \cdot \varepsilon / E_o \geq 1 \tag{11}$$

In order to reduce the influence of Earth perturbation differences in inclination angles and better reflect the relationship between orbital feasible domains and propulsion parameters, the orbital parameters in the equatorial plane are calculated. Side-to-cross-section ratios are (2:1, 3:1, 4:1, 5:1), thrust-to-power ratios are (25, 50, 75, 100) μN/W, and drag coefficients are (1, 2, 3, 4) (Figures 5–7, in which "P" means periapsis, and "A" means apoapsis).

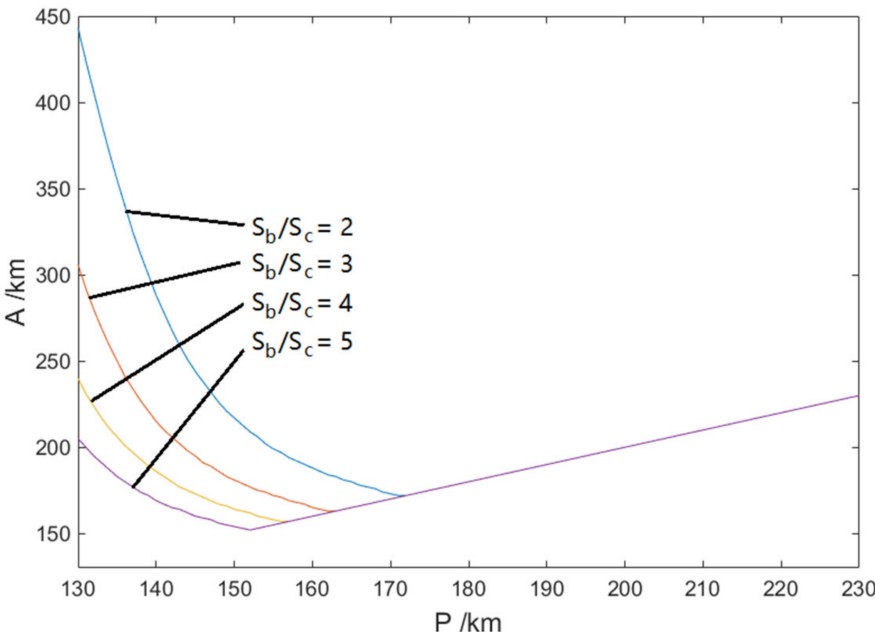

**Figure 5.** The feasible domain of elliptical orbit parameters decided by the side-to-cross-section ratio.

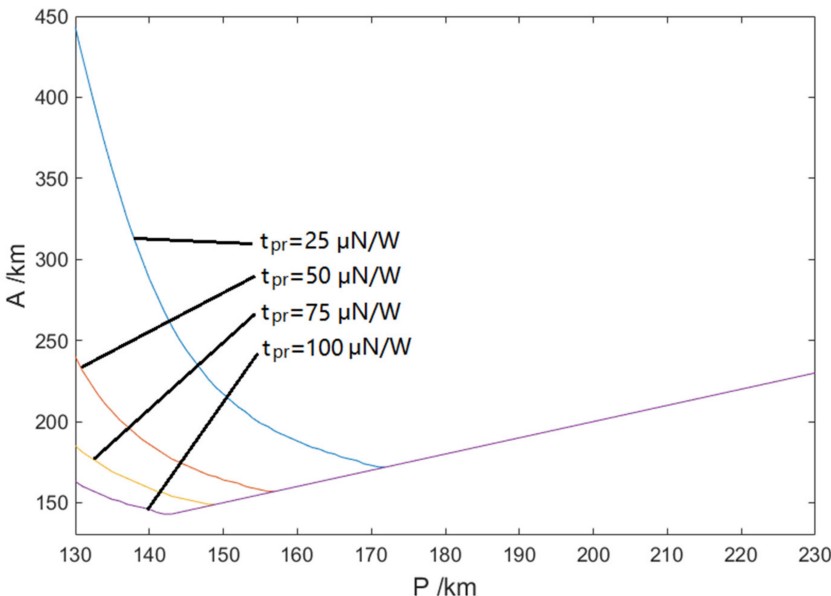

**Figure 6.** The feasible domain of elliptical orbit parameters decided by the thrust-to-power ratio.

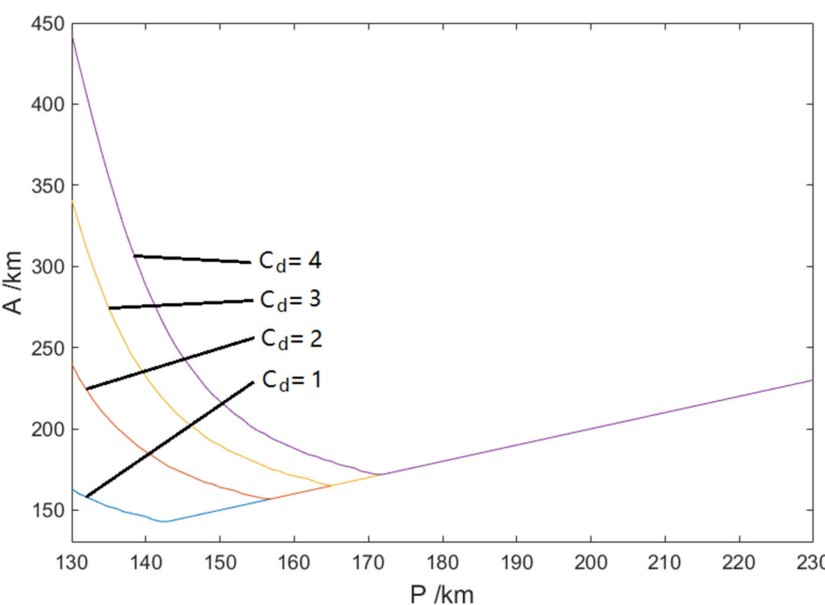

**Figure 7.** The feasible domain of elliptical orbit parameters decided by the drag coefficient.

The results show that these three parameters (side-to-cross-section ratio, power thrust ratio, and drag coefficient) all significantly affect the feasible domain of the orbit.

It is worth noting that Figures 5–7 show the situation where the power proportion of the propulsion system $\varepsilon$ is 100%, without considering the payload power supply and design redundancy, so the feasible domain will be smaller. In this case, Equation (11) should be rewritten as

$$E_i/E_o \geq 1 + \gamma, \tag{12}$$

where $\gamma$ is the payload power proportion considering the redundancy factor, which equals $1/\varepsilon - 1$.

The relationship between the orbital feasible domain and the payload power proportion $\gamma$ is shown in Figure 8 (the side-to-cross-section ratio is 4, the thrust-to-power ratio is 50 $\mu$N/W, and the drag coefficient is 1.5). Therefore, the feasible domain is further limited.

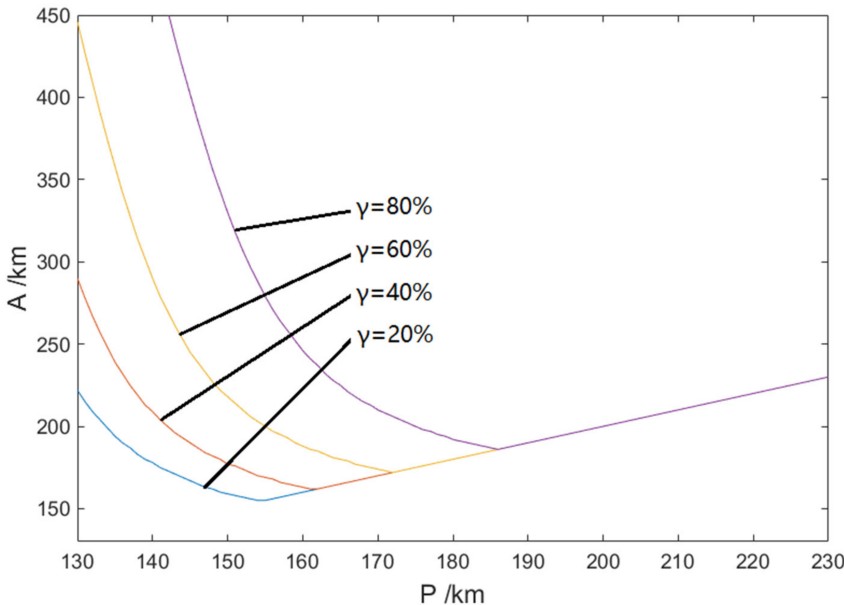

**Figure 8.** The feasible domain of elliptical orbit parameters decided by payload power proportion.

*3.2. Control Method of Elliptic Orbit in VLEO Space*

Although theoretical calculations show that a specific VLEO elliptical orbit can be maintained under proper propulsion parameters, the actual situation is much more complicated.

On the one hand, the actual atmospheric drag is not a predicted value decided by $\rho(h(t))$ function, i.e.,

$$F_{f\prime} = F_f + \Delta f = \frac{1}{2}\rho v^2 C_d(b) S_c + \Delta f, \tag{13}$$

in which $\Delta f$ is the unpredictable deviation value due to actual deviations from $\rho$. This makes it difficult for thrust to balance the actual in real-time according to Equation (4).

On the other hand, considering the thruster control strategy and system lifetime, the thrust of the thruster should also avoid long-term continuous changes.

Therefore, it is necessary to design a suitable orbit control strategy to ensure the long-term stable flight of the ABEP spacecraft in the target elliptical orbit. The control strategy based on orbital altitude deviation is first considered. Using the on–off control (Bang-Bang control) method, once the actual orbit perigee or apogee deviates from the set value, the thruster is activated for orbit correction.

We used the calculation package from the open-source General Mission Analysis Tool (GMAT) [37] in the simulation. A 150–250 km elliptical orbit is taken (selected from the feasible domain of Figure 8) with a thrust-to-power ratio of 50 μN/W, a drag coefficient of 1.5, a slenderness ratio of 4, and $\gamma$ = 40%. When the actual orbit deviates 5 km from the target orbit (i.e., <145 km/>155 km at perigee or <245 km/>255 km at apogee), the thruster is activated for orbit correction with a rated thrust $F_{t0}$. At this time, the control criterion is:

$$\begin{aligned} Flag &= \begin{cases} 1\,, A < A_{\min} \text{ or } A > A_{\max} \text{ or } P < P_{\min} \text{ or } P > P_{\max} \\ 0, others \end{cases} \\ F_t &= \begin{cases} F_{t0}, \; Flag = 1 \\ 0, \; Flag = 0 \end{cases} \end{aligned} \tag{14}$$

Figure 9 shows the trajectory height of the simulation case without thrust, with a rapid decay in orbital altitude and re-entry into the atmosphere in about 10 days. Figure 10 shows the flight altitude curves (a, b) and control curves (c, d) using the on–off control strategy based on altitude deviation which is described above. It can be seen that the spacecraft can maintain an elliptical orbital flight of 150–250 km stably for a long time. However, as can be seen from

the flight curve (Figure 11), the perigee argument $\omega$ of the orbit has been drifting, and the variation period of the argument of perigee is about 100 days, as shown in Figure 12.

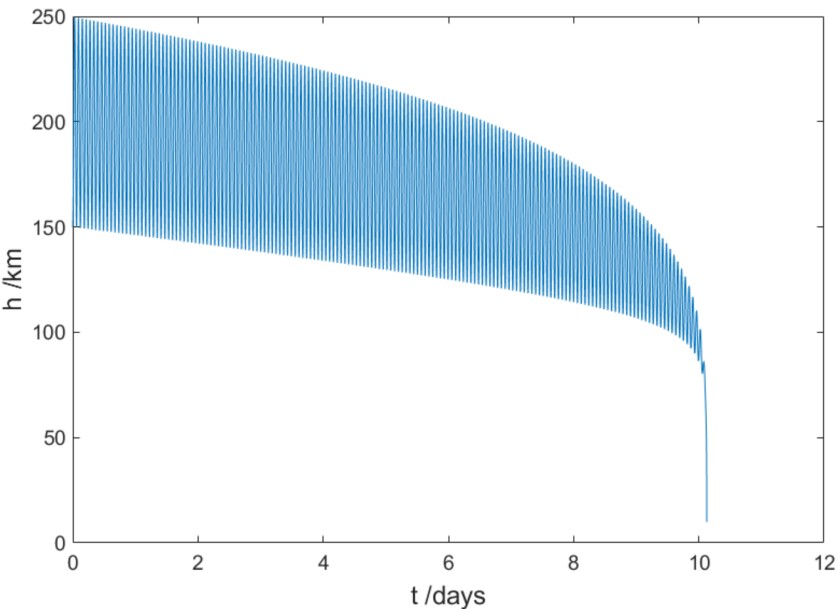

**Figure 9.** The trajectory height of the simulation case without thrust.

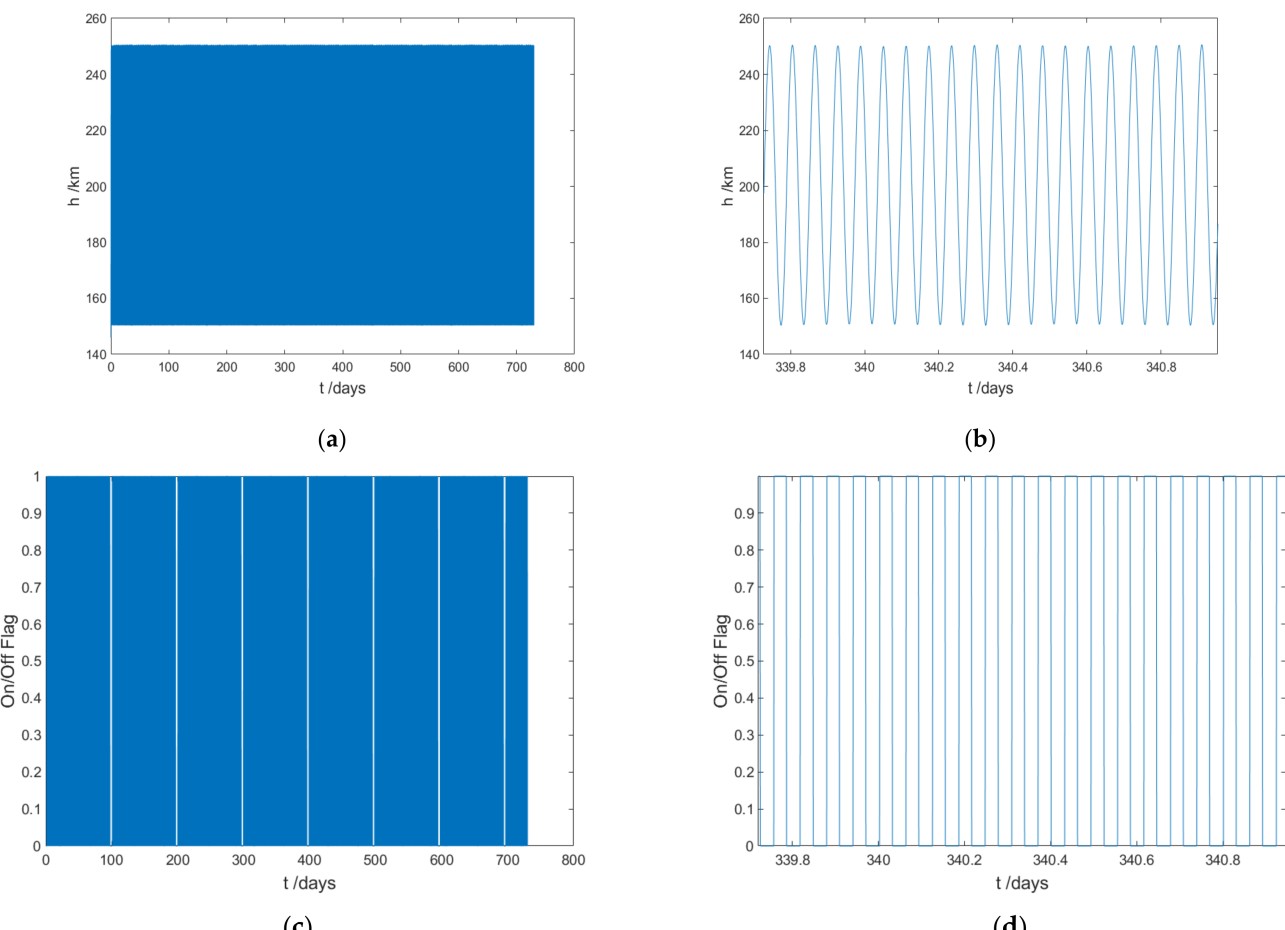

**Figure 10.** The mission profile of the simulation case using the on–off control method: (**a**,**b**) flight altitude curves; (**c**,**d**) control curves.

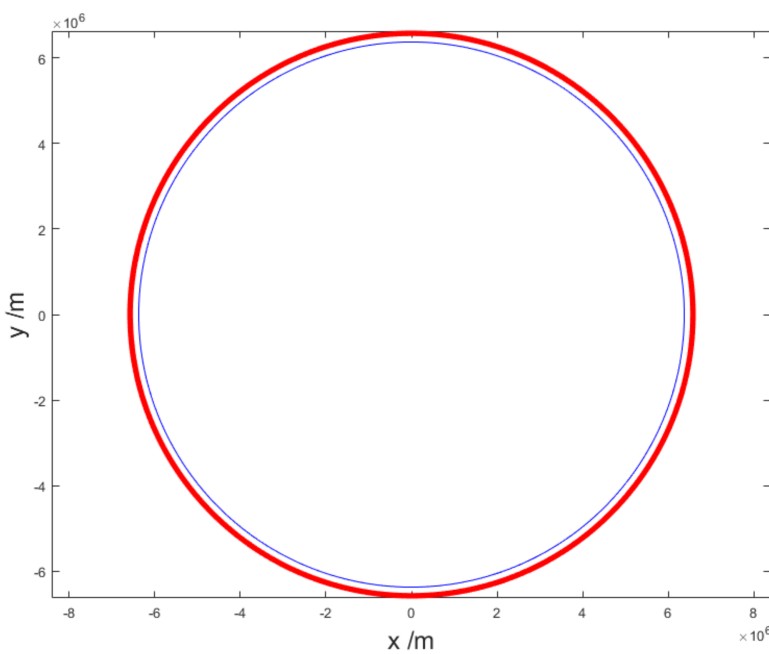

**Figure 11.** The flight trajectory under the on-off control method.

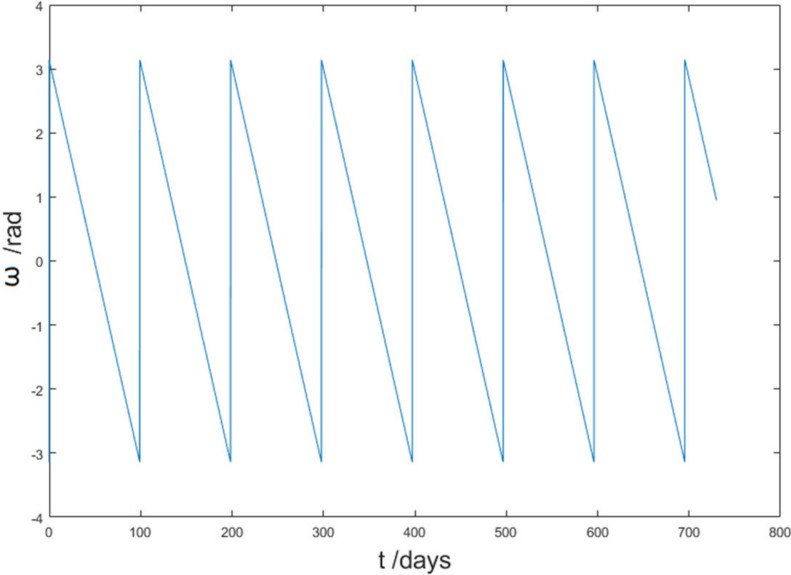

**Figure 12.** The argument of perigee under the on-off control method.

To avoid perigee drift, a half-period on–off control method with constant additional thrust can be adopted. This method adds a small constant extra thrust $F_{tc}$ (6% of the rated thrust, in order to balance the sustained resistance during the whole flight) to the control method described above, while the on–off control is performed only during half of the orbital period on the perigee side (Equation (15)). At this time, the height and argument of perigee of the elliptical orbit can be kept (as shown in Figures 13 and 14), that is, the orbit maintenance is achieved. This method requires the thruster to be switched on and off more often than the first method.

$$Flag\prime = \begin{cases} 1 \ , (M \in [-\frac{\pi}{2}, \frac{\pi}{2}]) \& (A < A_{\min} \ \text{or} \ A > A_{\max} \ \text{or} \ P < P_{\min} \ \text{or} \ P > P_{\max}) \\ 0, others \end{cases}$$

$$F_t = \begin{cases} F_{t0}, \ Flag\prime = 1 \\ F_{tc}, \ Flag\prime = 0 \end{cases} \tag{15}$$

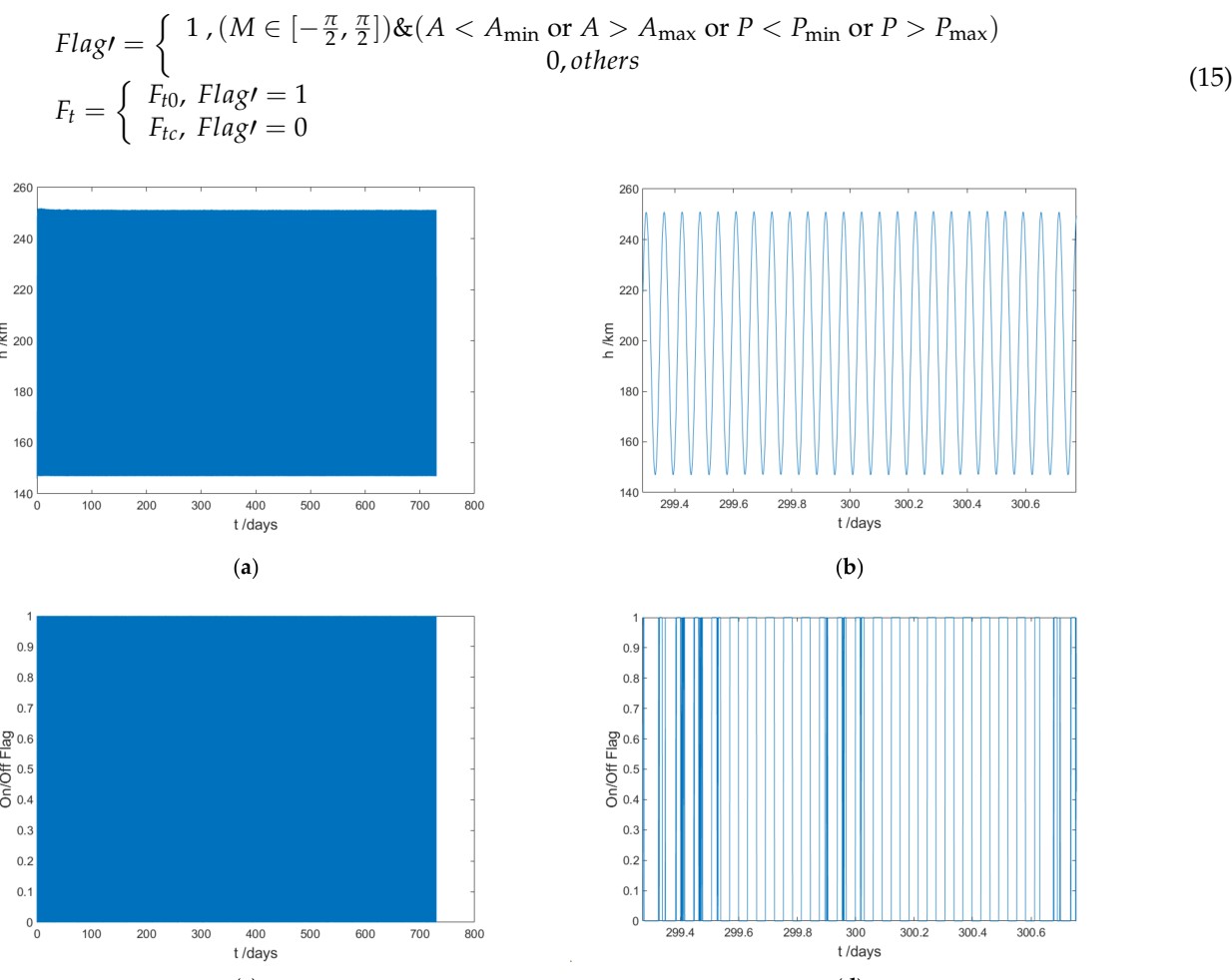

**Figure 13.** Mission profile of the simulation case using half-period on–off control method: (**a**,**b**) flight altitude curves; (**c**,**d**) control curves.

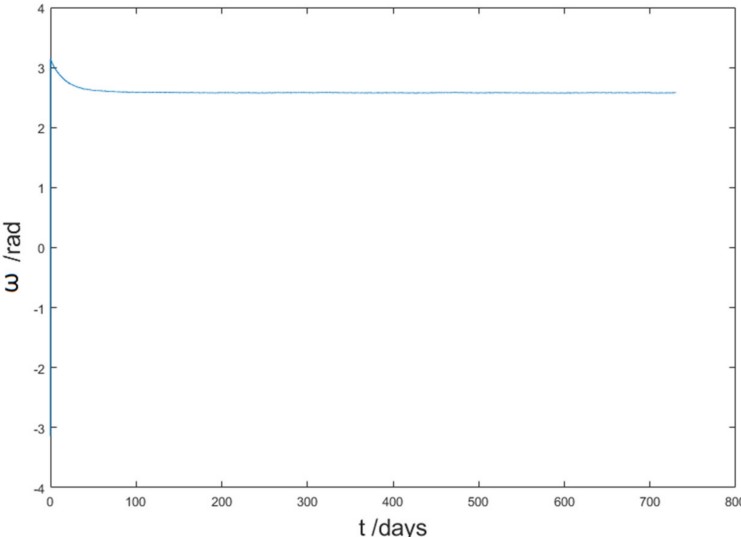

**Figure 14.** The argument of perigee under half-period on-off control method.

When the resistance deviation value $\Delta f$ is small, the effectiveness of the above two on–off control methods will not be affected. When the drag deviation value is large, the thrust of the spacecraft may not be able to balance the large drag generated for a period of time, which may cause control failure. Therefore, it is necessary to obtain more accurate and reliable data on the physical properties of the upper atmosphere. Pre-in-situ detection can be carried out by launching high-accuracy test satellites to determine the design parameters and control scheme of long-term ABEP satellites in orbit in the future.

## 4. Discussion and Conclusions

With the development of ABEP technology, the long-term on-orbit flight can be maintained in the VLEO space below 200 km in the near future. According to the analysis in this paper, the spacecraft can stay in an elliptical orbit with a perigee of about 150 km under the existing technical conditions. The main conclusions drawn by this research are listed below:

(1) The main constraints of ABEP spacecraft during VLEO flight were studied based on the normalized thrust-to-drag ratio and the normalized energy balance parameter (the power ratio). The drag coefficient, the effective specific impulse, the slenderness ratio, and the thrust-to-power ratio were analyzed as the main parameters affecting the feasible range of the orbit of ABEP spacecraft. The energy balance is the key constraint for low VLEO orbits, which is determined by the drag coefficient, slenderness ratio, and thrust-to-power ratio. Under the existing technical conditions, the lowest circular orbit (along the terminator) of the spacecraft can reach about 170 km (the drag coefficient is 2, the slenderness ratio is 4, and the thrust-to-power ratio of the Hall thruster is 50 µN/W). These three parameters are expected to be further optimized through future studies to extend the lower limit of the feasible domain of the ABEP spacecraft orbit.

(2) To reach the orbital flight altitude of 150 km, an elliptical orbit flight scheme in VLEO space for ABEP spacecraft is proposed. The feasible domain of orbital parameters is analyzed based on the slenderness ratio, the thrust-to-power ratio, and the drag coefficient. Payload power redundancy also limits further reductions in the feasible height of elliptical orbits, so the development of additional energy supply technologies (such as wireless power beaming) will extend the lower orbit limit of VLEO missions.

(3) Based on the on–off control method widely used in electric propulsion, two control methods are proposed to maintain the 150–250 km elliptical orbit. Among them, the simple on–off control method based on orbital height deviation can keep the elliptical orbital height stable, but there is a significant drift in the orbital perigee argument. The half-period on–off control method with constant additional thrust is proposed, which can stabilize the argument of perigee while ensuring the height of the elliptical orbit, and this method is expected to be applied in future elliptical orbit missions in VLEO space.

**Author Contributions:** Conceptualization, W.L.; methodology, software, and writing—original draft preparation Y.Y.; formal analysis and writing—review and editing, J.G.; investigation and validation, G.F. All authors have read and agreed to the published version of the manuscript.

**Funding:** This research was funded by NSFC: No. 12275019.

**Data Availability Statement:** The data presented in this study are available on request from the corresponding author.

**Acknowledgments:** This research is supported by The Key Scientific Research Program of Chinese Academy of Sciences (program number: KGFZD-145-23-08). This research is also supported by the Youth Innovation Promotion Association of CAS.

**Conflicts of Interest:** The authors declare no conflict of interest.

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
