# Peer review of "Elliptical Orbit Design Based on Air-Breathing Electric Propulsion Technology in Very-Low Earth Orbit Space"

_aerospace, doi:10.3390/aerospace10100899_

Round 1
Reviewer 1 Report
Review of manuscript
MDPI Aerospace 2648089
“Elliptical orbit design based on air-breathing electric propulsion technology in Very-low Earth orbit space”
By Yuxian Yue, Jinyue Geng, Guanhua Feng, Wenhao Li
The subject of this paper is very actual and interesting for the space community, e.g. can electric propulsion combined with using atmospheric air capture and its use as propellant give enough thrust for missions in very low orbits.
My first observation was the English is not adequate and errors made reading not satisfactory. A better final proof reading before submission should have been done. E.g., very often VLEO is confused with VELO.
Atmospheric drag depends on air density, and here I would spend more details on density variability. This can be seen in other papers. Just mentioning the use of the MSISE90 model is not enough.
What I miss too is some details about practical solutions. Doing RAM-EP with air compression and thruster operation in real life is very challenging, and this should be mentioned more.
Some references should be added and mentioned, e.g. the review article “A review of air-breathing electric propulsion: from mission studies to technology verification” by Tommaso Andreussi, Eugenio Ferrato and Vittorio Giannetti
In summary, I think that it is worth continuing with the publication procedure, but a major editing is necessary.
My first observation was the English is not adequate and errors made reading not satisfactory. A better final proof reading before submission should have been done. E.g., very often VLEO is confused with VELO.
There are many errors, and for me it does not make sense to correct them.
Reviewer 2 Report
See attachment

Minor edits recommended.
Round 2
Reviewer 1 Report
Please extend a bit more the abstract according to the conclusion section.
Please do a final run concerning English quality (native speaker?).
Author Response
Thank you for your advice.
The article has been revised (the abstract has been extended according to conclusions, and the English quality has been improved).
All changes have been marked red in the revised manuscript (version: r2).